# Atherogenic Lipoproteins for the Statin Residual Cardiovascular Disease Risk

**DOI:** 10.3390/ijms232113499

**Published:** 2022-11-04

**Authors:** Hidekatsu Yanai, Hiroki Adachi, Mariko Hakoshima, Hisayuki Katsuyama

**Affiliations:** Department of Diabetes, Endocrinology and Metabolism, National Center for Global Health and Medicine Kohnodai Hospital, 1-7-1 Kohnodai, Chiba 272-8516, Japan

**Keywords:** cardiovascular disease, lipoprotein (a), malondialdehyde-modified LDL, remnant lipoproteins, small-dense LDL, statins

## Abstract

Randomized controlled trials (RCTs) show that decreases in low-density lipoprotein cholesterol (LDL-C) by the use of statins cause a significant reduction in the development of cardiovascular disease (CVD). However, one of our previous studies showed that, among eight RCTs that investigated the effect of statins vs. a placebo on CVD development, 56–79% of patients had residual CVD risk after the trials. In three RCTs that investigated the effect of a high dose vs. a usual dose of statins on CVD development, 78–87% of patients in the high-dose statin arms still had residual CVD risk. The risk of CVD development remains even when statins are used to strongly reduce LDL-C, and this type of risk is now regarded as statin residual CVD risk. Our study shows that elevated triglyceride (TG) levels, reduced high-density lipoprotein cholesterol (HDL-C), and the existence of obesity/insulin resistance and diabetes may be important metabolic factors that determine statin residual CVD risk. Here, we discuss atherogenic lipoproteins that were not investigated in such RCTs, such as lipoprotein (a) (Lp(a)), remnant lipoproteins, malondialdehyde-modified LDL (MDA-LDL), and small-dense LDL (Sd-LDL). Lp(a) is under strong genetic control by apolipoprotein (a), which is an LPA gene locus. Variations in the LPA gene account for 91% of the variability in the plasma concentration of Lp(a). A meta-analysis showed that genetic variations at the LPA locus are associated with CVD events during statin therapy, independent of the extent of LDL lowering, providing support for exploring strategies targeting circulating concentrations of Lp(a) to reduce CVD events in patients receiving statins. Remnant lipoproteins and small-dense LDL are highly associated with high TG levels, low HDL-C, and obesity/insulin resistance. MDA-LDL is a representative form of oxidized LDL and plays important roles in the formation and development of the primary lesions of atherosclerosis. MDA-LDL levels were higher in CVD patients and diabetic patients than in the control subjects. Furthermore, we demonstrated the atherogenic properties of such lipoproteins and their association with CVD as well as therapeutic approaches.

## 1. Introduction

Epidemiological studies show that an elevation of serum low-density lipoprotein cholesterol (LDL-C) increases the incidence and mortality of cardiovascular disease (CVD), such as coronary heart disease (CHD). Randomized controlled trials (RCTs) of cholesterol lowering in patients with and without CHD showed that decreasing LDL-C by using statins caused a significant reduction in CVD [1,2,3,4,5,6,7,8]. Furthermore, intense LDL-C-lowering therapy using a high dose of strong statins reduced CV events [9,10,11].

However, one of our studies showed that, among eight RCTs that investigated the effect of statins vs. a placebo on CVD development, 56–79% of patients had a residual CVD risk after trials [12]. In three RCTs that investigated the effect of a high dose vs. a usual dose of statins on CVD development, 78–87% of patients in the high-dose statin arms still had a residual CVD risk after trials [12]. The risk of CVD development remains even when statins are used to strongly reduce LDL-C, and such a risk is regarded as a statin residual CVD risk. Our analysis of the characteristics of patients in such RCTs suggested that elevated triglyceride (TG) levels, reduced high-density lipoprotein cholesterol (HDL-C), and the existence of obesity/insulin resistance and diabetes may be important metabolic factors that determine statin residual CVD risk.

Here, we discuss atherogenic lipoproteins that were not investigated in such RCTs, such as lipoprotein (a) (Lp(a)), remnant lipoprotein, malondialdehyde-modified LDL (MDA-LDL), and small-dense LDL (Sd-LDL).

## 2. Lipoprotein (a) (Lp(a))

Lp(a) was initially described by Kare Berg in 1963 [13]; it has a complex structure that combines a cholesterol-rich LDL-like particle and apolipoprotein (a) (Apo(a)), a glycoprotein of variable size. Lp(a) represents a genetically transmitted class of serum LDL that has apo B-100 linked by a disulfide bridge to Apo(a) [14]. Apo(a) shares a high degree of sequence homology with plasminogen (75–94%) [15,16]. Plasminogen is a plasma serine protease zymogen that consists of five homologous and tandemly repeated domains called kringles and a trypsin-like protease domain [15]. The amino-terminal sequence obtained for Apo(a) is homologous to kringle IV of plasminogen. Lp(a) is under strong genetic control by the Apo(a) gene, an LPA gene locus [17]. Genetic variants, including a highly polymorphic copy number variation of kringle IV repeats at this locus, have a pronounced influence on Lp(a) concentrations. Variations in the LPA gene, which encodes Apo(a), account for 91% of the variability in the plasma concentration of Lp(a) [18] and explain differences in circulating concentrations between different populations.

### 2.1. Atherogenic and Thrombogenic Properties of Lp(a)

The atherogenic and thrombogenic properties of Lp(a) are shown in Figure 1. Lp(a) has been found within the intima of human arteries [19], indicating that Lp(a) can enter and accumulate in the intima of arteries. Lp(a) may be preferentially trapped in injured arterial intima due to the greater capacity of Lp(a) to bind to fibrin or glycosaminoglycans, which enhances the adhesion of Lp(a) to endothelial cells (Figure 1(1)) [19]. Subjects with elevated Lp(a) have increased arterial inflammation and enhanced peripheral blood mononuclear cell trafficking to the arterial wall compared with subjects with normal Lp(a) (Figure 1(2)) [15]. Lp(a) contains oxidized phospholipids (OxPLs) and augments the proinflammatory response in monocytes derived from healthy control subjects (Figure 1(3)) [20]. Lp(a) can be taken up by monocyte-derived macrophages that induce foam cell formation (Figure 1(4)), and Lp(a) contributes to smooth muscle cell proliferation and migration (Figure 1(5)) [19,21]. OxPLs colocalize with Apo(a)-Lp(a) in arterial and aortic valve lesions and directly participate in the pathogenesis of these disorders by promoting endothelial dysfunction (Figure 1(6)), lipid deposition (Figure 1(4)), inflammation (Figure 1(3)), and osteogenic differentiation (Figure 1(7)), leading to calcification [22].

The structural similarity of Apo(a) and plasminogen is a key factor explaining the causal relationship between Lp(a) and atherothrombosis. Apo(a) may be of pathophysiological relevance by virtue of binding to fibrin and cell surfaces, thus inhibiting plasmin formation [23]. Plasminogen and plasmin tether to cell surfaces through a ubiquitously expressed and structurally quite dissimilar family of proteins, as well as some nonproteins, collectively referred to as plasminogen receptors. Plasminogen receptors have been shown to facilitate plasminogen activation to plasmin and to protect bound plasmin from inactivation by inhibitors. Lp(a) binds to fibrin surfaces and to a variety of plasminogen receptors [24,25], thereby competing with plasminogen and inhibiting its activation (Figure 1(8)) [26]. In fibrinolysis, plasminogen is rate-limiting for plasmin formation on fibrin and the surfaces of platelets and cells of the injured vascular wall [27]. Reduced fibrinolysis, with an accumulation of Lp(a) at sites of vascular injury, may represent the link between elevated Lp(a) concentrations and the development of atherothrombosis (Figure 1(9)). A study that investigated the binding of different recombinant Apo(a) isoforms showed that the prothrombotic action of Lp(a) may be in part mediated by the modulation of the platelet function through the interaction of its Apo(a) subunit with a specific receptor at the platelet surface [28], indicating that the binding of Lp(a) to platelets induces platelet activation (Figure 1(10)). Platelet activation is highly associated with the progression of atherosclerosis as well as thrombosis formation [29]. Lp(a) regulates the endothelial cell synthesis of a major fibrinolytic protein, plasminogen activator inhibitor-1 (PAI-1). In cultured human endothelial cells, Lp(a) enhanced PAI-1 antigen, activity, and steady-state mRNA levels without altering tissue plasminogen activator (tPA) activity or mRNA transcript levels (Figure 1(11)), suggesting that Lp(a) stimulates the expression of PAI-1 independently of tPA [30].

### 2.2. Associations of Lp(a) with Various Diseases

#### 2.2.1. CHD

Approximately 20% of the global population (1.4 billion people) has elevated levels of Lp(a), which are associated with higher CV risk, though the threshold for determining “high risk” is debated. In a meta-analysis, serum Lp(a) concentrations were higher in subjects who later developed ischemic heart disease (IHD) cases than in controls, providing evidence in support of a causal role for Lp(a) in the development of atherosclerotic CVD [31]. In a meta-analysis of prospective studies with at least 1 year of follow-up published before 2000, a comparison of individuals in the top third of baseline plasma Lp(a) measurements with those in the bottom third in each study yielded a combined risk ratio of 1.6 (95% CI: 1.4–1.8; *p* < 0.00001), demonstrating a clear association between Lp(a) and IHD [32].

Clarke et al., used a novel gene chip containing 48,742 single-nucleotide polymorphisms (SNPs) in 2100 candidate genes to test for associations in 3145 case subjects with CHD and 3352 control subjects [33]. Three chromosomal regions (6q26-27, 9p21, and 1p13) were strongly associated with the risk of CHD. A meta-analysis showed that, with a genotype score involving both LPA SNPs, the odds ratios (OR) for CHD were 1.51 (95% CI: 1.38–1.66) for one variant and 2.57 (95% CI: 1.80–3.67) for two or more variants. They identified LPA variants that were strongly associated with both an increased level of Lp(a) and an increased risk of CHD, supporting a causal role of Lp(a) in CHD. In a study assessing the association of Apo(a) isoforms with CV risk, people with smaller Apo(a) isoforms had an approximately two-fold higher risk of CHD than those with larger proteins [34].

Elevated Lp(a) is recognized as a risk factor for incident CVD in the general population and established CVD patients. However, there are conflicting findings on the prognostic utility of elevated Lp(a) in patients with CHD. Recently, a meta-analysis was performed to evaluate the prognostic value of elevated Lp(a) in CHD patients [35]. The meta-analysis indicated that elevated Lp(a) was independently associated with an increased risk of cardiac events (cardiac death and acute coronary syndrome (ACS)) (relative risk (RR): 1.78; 95% CI: 1.31–2.42) and CV events (death, stroke, ACS, or coronary revascularization) (RR: 1.29; 95% CI: 1.17–1.42) in CHD patients. However, an elevated Lp(a) level was not significantly associated with an increased risk of cardiovascular mortality (RR: 1.43; 95% CI: 0.94–2.18) or all-cause mortality (RR: 1.35; 95% CI: 0.93–1.95).

According to the guidelines by the American College of Cardiology (ACC) and American Heart Association (AHA), an Lp(a) level of less than 30 mg/dL indicates no significant risk of CVD, and an Lp(a) level greater than 50 mg/dL is regarded as a CVD-risk-enhancing factor [36]. The European Society of Cardiology (ESC) and the European Atherosclerosis Society (EAS) Guidelines have also mentioned that an Lp(a) level greater than 50 mg/dL indicates a high risk of CVD [37].

#### 2.2.2. Stroke

A meta-analysis using articles on Lp(a) and cerebrovascular disease was performed [38]. In case-control studies (*n* = 23), the unadjusted mean Lp(a) was higher in stroke patients (standardized mean difference (SMD): 0.39; 95% CI: 0.23–0.54) and was more frequently abnormally elevated (odds ratio (OR): 2.39; 95% CI: 1.57–3.63). In prospective cohort studies (*n* = 5), incident stroke was more frequent in patients in the highest tertile of Lp(a) distribution compared with the lowest tertile of Lp(a) (RR: 1.22; 95% CI: 1.04–1.43). This meta-analysis suggests that elevated Lp(a) is a risk factor for incident stroke. In another meta-analysis comparing high and low Lp(a) levels, the pooled estimated OR was 1.41 (95% CI: 1.26–1.57) for case-control studies (*n* = 11), and the pooled estimated RR was 1.29 (95% CI: 1.06–1.58) for prospective studies (*n* = 9) [39]. Study populations with a mean age of ≤55 years had an increased RR compared to older study populations. Elevated Lp(a) is an independent risk factor for ischemic stroke and may be especially relevant for young stroke patients.

The association of Lp(a) levels with the risk of stroke and its subtypes was investigated [40]. A significant association between increased levels of Lp(a) and the risk of ischemic stroke as compared to control subjects was observed (SMD: 0.76; 95% CI: 0.53–0.99). Lp(a) levels were also found to be significantly associated with the risk of the large artery atherosclerosis subtype of ischemic stroke (SMD: 0.68; 95% CI: 0.01–1.34) and significantly associated with the risk of intracerebral hemorrhage (SMD: 0.65; 95% CI: 0.13–1.17) as compared to controls.

#### 2.2.3. Peripheral Artery Disease (PAD)

The relevance of Lp(a) concentrations and Apo(a) phenotypes in PAD has been investigated in few studies. Laschkolnig et al., analyzed this association in three independent cohorts and employed a Mendelian randomization approach using instrumental variable regression [41]. Analyses of three independent populations showed significant associations between PAD and Lp(a) concentrations, Apo(a) phenotypes, and one SNP in the LPA gene (rs10455872).

The effects of variants in LPA on vascular diseases with different atherosclerotic and thrombotic components were investigated [42]. LPA score was associated with the large artery atherosclerosis (OR: 1.27; *p* < 0.001), PAD (OR: 1.47; *p* < 0.001), and abdominal aortic aneurysm (OR: 1.23; *p* < 0.001) subtypes of ischemic stroke, but not with the cardio-embolism (OR: 1.03; *p* = 0.69) or small vessel disease (OR: 1.06; *p* = 0.52) subtypes.

Although the genetically determined Apo(a) phenotype and SNP in the LPA gene are associated with the development of PAD, the contribution of Lp(a) to the development of PAD remains to be elucidated.

#### 2.2.4. Venous Thromboembolism

The published data on the association between high Lp(a) levels and venous thromboembolism were systematically examined [43]. Six case-control studies were included, incorporating 1826 cases of venous thromboembolism and 1074 controls. Lp(a) levels >300 mg/L were significantly associated with venous thromboembolism (OR: 1.87; 95% CI: 1.51–2.30; *p* < 0.0001). This meta-analysis shows a significant association between high Lp(a) levels and the occurrence of venous thromboembolism in adults. Another systematic review of the literature to better clarify role of Lp(a) as a risk factor for venous thromboembolism was performed [44]. Fourteen studies for a total of more than 14,000 patients were ultimately included. Lp(a) was significantly associated with an increased risk of venous thromboembolism (OR: 1.56; 95% CI: 1.36–1.79). Venous thromboembolism patients had significantly higher Lp(a) values compared with controls (weighted MD (WMD): 14.46 mg/L; 95% CI: 12.14–16.78).

Paciullo et al., performed a systematic review and meta-analysis of the studies addressing the role of Lp(a) in retinal vein occlusion [45]. Lp(a) levels above normal limits were associated with retinal vein occlusion (OR: 2.38; 95% CI: 1.7–3.34), and patients with retinal vein occlusion had higher Lp(a) levels than controls (WMD: 13.4 mg/dL; 95% CI: 8.2–18.6).

#### 2.2.5. Aortic Valve Calcification and Stenosis

Limited information is available regarding genetic contributions to valvular calcification. A genome-wide association study revealed that genetic variation in the LPA locus, mediated by Lp(a) levels, is associated with aortic valve calcification across multiple ethnic groups and with incident clinical aortic stenosis [46]. In a meta-analysis including 4651 cases and 8231 controls, the CHD-associated allele at the LPA locus was associated with an increased risk of aortic valve stenosis (OR: 1.37; 95% CI: 1.24–1.52; *p* < 0.001), with a larger effect size in those without CHD (OR: 1.53; 95% CI: 1.31–1.79) compared to those with CHD (OR: 1.27; 95% CI: 1.12–1.45) [47].

In a systematic review and meta-analysis aimed at determining the association between plasma Lp(a) levels and aortic valve calcification [48], the pooled results showed that plasma Lp(a) levels ≥ 50 mg/dL were associated with a 1.76-fold increased risk of aortic valve calcification (RR: 1.76; 95% CI: 1.47–2.11), but Lp(a) levels ≥ 30 mg/dL were not observed to be significantly related to aortic valve calcification (RR: 1.28; 95% CI: 0.98–1.68).

#### 2.2.6. Diabetic Nephropathy

A meta-analysis showed that, compared to those with the lowest Lp(a) levels, patients with the highest Lp(a) levels had higher odds of diabetic nephropathy (OR: 1.63; 95% CI: 1.25–2.14; *p* < 0.001), suggesting that higher levels of serum Lp(a) in patients with type 2 diabetes are independently associated with the development of diabetic nephropathy [49].

### 2.3. Therapeutic Approaches to Lower Lp(a)

In 16,654 individuals from the EPIC-Norfolk prospective population study and in 9448 individuals from the Copenhagen City Heart Study (CCHS), parallel statistical analyses were performed [50]. Lp(a) and LDL-C were found to be independently associated with CVD risk. At LDL-C levels below <2.5 mmol/L (97 mg/dL), the risk associated with elevated Lp(a) attenuates in a primary prevention setting [50].

The variants of LPA have been shown to confer a significant increase in the risk for CHD [33,34]; the large artery atherosclerosis, PAD, and abdominal aortic aneurysm subtypes of ischemic stroke [42]; aortic valve calcification [46]; and aortic valve stenosis [47]. The current state-of-the-art therapeutic interventions tend to rely upon the useful apprehension of such data points. A meta-analysis showed that the variants of LPA are associated with CVD events during statin therapy independently of the extent of LDL lowering, suggesting a significant contribution of LPA variants to statin residual CVD risk [51].

#### 2.3.1. Diet and Exercise

An elevated level of Lp(a) is approximately 90% genetically regulated. Non-genetic factors that may influence Lp(a) levels, such as diet and physical activity, largely remain unknown. Results from studies on the association between Lp(a) levels and physical activity/exercise have been inconsistent [52].

#### 2.3.2. Statins

A meta-analysis showed that genetic variations at the LPA locus are associated with CHD during statin therapy, independent of the extent of LDL lowering, providing support for exploring strategies targeting circulating concentrations of Lp(a) to reduce CHD events in patients receiving statins [51]. In the individual-patient data meta-analysis of statin-treated patients, elevated baseline and on-statin Lp(a) levels showed an independent, approximately linear relation to CVD risk [53].

A subject-level meta-analysis including 5256 patients (1371 on a placebo and 3885 on statins) from six RCTs, three statin-vs.-placebo trials, and three statin-vs.-statin trials, with pre- and on-treatment (4–104 weeks) Lp(a) levels, was performed [54]. The mean percent change in Lp(a) from baseline ranged from 8.5% to 19.6% in the statin groups and from −0.4% to −2.3% in the placebo groups. The mean percent change of Lp(a) from baseline ranged from 11.6% to 20.4% in the pravastatin group and from 18.7% to 24.2% in the atorvastatin group. This meta-analysis reveals that statins significantly increase plasma Lp(a) levels. However, a meta-analysis including 12 RCTs with 23,605 participants showed that statins have no clinically significant effect on Lp(a) levels and that there is no significant difference in the effect on Lp(a) levels between different types and dosages of statins [55]. A recent meta-analysis of 39 studies (24,448 participants) showed that statin therapy does not lead to clinically important differences in Lp(a), compared to a placebo, in patients at risk for CVD [53]. Based on the results of these meta-analyses, we think that statin therapy will not change Lp(a)-associated CVD risk.

In an individual-patient data meta-analysis of statin-treated patients, associations of baseline and on-statin treatment Lp(a) with CVD risk were approximately linear, with an increased risk at Lp(a) values equal to or greater than 30 mg/dL for baseline lipoprotein(a) and equal to or greater than 50 mg/dL for on-statin Lp(a) [56].

#### 2.3.3. Niacin

A meta-analysis suggested a significant reduction in Lp(a) levels following extended-release (ER) niacin treatment (WMD: −22.90%; 95% CI: −27.32 to −18.48; *p* < 0.001) [57]. When the studies were categorized according to the administered dose, there was a comparable effect between the subsets of the studies with administered doses of <2000 mg/day (WMD: −21.85%; 95% CI: −30.61–−13.10; *p* < 0.001) and those with doses of ≥2000 mg/day (WMD: −23.21%; 95% CI: −28.41–−18.01; *p* < 0.001).

The HPS2-THRIVE (Heart Protection Study 2-Treatment of HDL to Reduce the Incidence of Vascular Events) study showed that the mean proportional reduction in Lp(a) with niacin–laropiprant was 31% but varied strongly with the size of the predominant Apo(a) isoform and was only 18% in the quintile with the highest baseline Lp(a) level and a low isoform size [58]. Estimates from genetic studies suggest that these Lp(a) reductions during the short term of the trial might yield proportional reductions in coronary risk of approximately 2% overall and 6% in the top quintile by Lp(a) level. The likely benefits of niacin–laropiprant for coronary risk through Lp(a) lowering are small. Novel therapies that reduce high Lp(a) levels by at least 40% may be needed to produce worthwhile benefits in people at the highest risk due to Lp(a).

#### 2.3.4. Fibrates and Ezetimibe

A meta-analysis showed that the Lp(a)-reducing effect of fibrate monotherapy was modest and non-significant (WMD: −1.76 mg/dL; 95% CI: −5.44–−1.92; *p* = 0.349) [59].

Ezetimibe monotherapy (10 mg/day) showed a small (7.06%) but statistically significant reduction in the plasma levels of Lp(a) in patients with primary hypercholesterolemia [60]. According to the current literature, this magnitude of reduction seems to have no clinical relevance. Another meta-analysis suggested that ezetimibe treatment either alone or in combination with a statin does not affect plasma Lp(a) levels [61].

#### 2.3.5. Hormone Replacement Therapy (HRT)

A systematic review using 248 studies provided information on the effects of 42 different HRT regimens [62]. In 41 studies of 20 different formulations, HRT generally lowered Lp(a). A meta-analysis that studied the effect of HRT on components of metabolic syndrome in postmenopausal women showed that HRT reduced Lp(a) levels (−25.0%; 95% CI: −32.9–−17.1) [63]. Another meta-analysis examining 24 studies showed that HRT caused a significant reduction in Lp(a) concentrations compared with a placebo or no treatment (mean relative difference: −20.35%; 95% CI: −25.33–−15.37; *p* < 0.0001) [64].

#### 2.3.6. Proprotein Convertase Subtilisin/Kexin Type 9 (PCSK9) Inhibitors

Evolocumab, a monoclonal antibody to PCSK9, decreases Lp(a). A pooled analysis of Lp(a) and LDL-C in 3278 patients from 10 clinical trials was conducted. Within each parent study, biweekly and monthly doses of evolocumab statistically significantly reduced Lp(a) at week 12 versus the control (*p* < 0.001 within each study), and the percent reductions were 24.7% and 21.7%, respectively [65]. A meta-analysis of the effect of PCSK9 inhibitors on circulating Lp(a) levels showed that PCSK9 inhibitors showed significant efficacy in reducing Lp(a) (−21.9%; 95% CI: −24.3–−19.5), irrespective of PCSK9 inhibitor types, treatment duration, participant characteristics, treatment methods, differences in the control treatment, baseline Lp(a) levels, and test methods [66]. The greatest reduction was achieved with 150 mg of alirocumab administered biweekly (−24.6%; 95% CI: −28.0–−21.2) and 140 mg of evolocumab administered monthly (−26.8%; 95% CI: −31.6–−21.9). In another meta-analysis, PCSK9 inhibitors altered Lp(a) levels by −26.7% (95% CI: −29.5%–−23.9%) [67].

PCSK9 inhibitors consistently reduce Lp(a) by 20–30%. Although it is not possible to conclude that PCSK9 inhibitors specifically lower Lp(a)-attributable CVD risk, patients with elevated Lp(a) could derive an incremental benefit from PCSK9 inhibitor therapy.

#### 2.3.7. Lipoprotein Apheresis

A single apheresis session can acutely decrease Lp(a) by approximately 60–75%, and apheresis performed weekly or biweekly results in considerably decreased mean interval concentrations (approximately 25–40% reduction) [68]. There are no RCTs showing a reduction in CVD by treating high Lp(a) levels with lipoprotein apheresis.

#### 2.3.8. Antisense Oligonucleotides (ASOs) and Small Interfering RNA (siRNA) Targeting Lp(a)

N-acetylgalactosamine-conjugated gene-silencing therapeutics, such as siRNA and antisense oligonucleotide-targeting LPA, are ideally suited for this application, offering a highly tissue- and target-transcript-specific approach with the potential for safe and durable Lp(a) lowering with as few as three or four doses per year [69].

To lower Lp(a), two antisense oligonucleotides (ASOs) have been developed: one targeting apo B and one targeting Apo(a) [70]. Mipomersen is an antisense oligonucleotide that targets apo B and has been shown to lower Lp(a) by 20–50% in Phase 3 studies. AKCEA-APO(a)-LRx is the most recent antisense oligonucleotide targeting Apo(a), thereby uniquely targeting Lp(a). It has been tested in a Phase 2 study and has been shown to lower Lp(a) levels by 50–80%. Such treatment seems promising; however, no improvement in CV risk has yet been shown.

## 3. Remnant Lipoproteins

Remnant lipoproteins are intermediate metabolic lipoproteins produced during the metabolism of TG-rich lipoproteins such as CM and VLDL. Evidence shows that, in humans and experimental animals, CM remnants and LDL are taken up by arterial cells [71]. Research directions that may contribute to the evaluation of CM remnants as a risk factor for atherogenesis are now being discussed. After the introduction of statins, clinical emphasis first focused on LDL-C lowering and subsequently on the potential for raising HDL-C, with less focus on lowering TG levels. The renewed interest in TGs has been driven by epidemiological and genetic evidence supporting raised TG levels, remnant cholesterol, and TG-rich lipoproteins as possible additional causes of CVD [72].

Recent epidemiologic studies have revealed that hypertriglyceridemia is associated with atherosclerosis, independent of other coronary risk factors. Atherosclerotic diseases with high TG levels can be found in patients with familial combined hyperlipidemia, type 2 diabetes, and metabolic syndrome, in which remnant lipoproteins accumulate in the circulating blood [73]. Dysbetalipoproteinemia (remnant clearance disease and Fredrickson type III hyperlipidemia) is an uncommon dyslipoproteinemia related to mutations in apolipoprotein E that disrupt the clearance of remnants of TG-rich lipoproteins [74]. Elevated remnant lipoproteins are also observed in patients with chronic kidney disease (CKD) [75].

A new homogenous assay for remnant lipoprotein cholesterol (RemL-C) has been developed as an alternative to remnant-like particle cholesterol (RLP-C), an immunoseparation assay widely used for the measurement of remnant lipoproteins cholesterol. We previously evaluated the correlations and data validation between the two assays and investigated the characteristics of remnant lipoproteins obtained by the two methods (RLP-C and RemL-C) and their relationships with IDL-C as determined by our developed high-performance liquid chromatography (HPLC) method [76]. A positive, significant correlation was found between the two methods (r, 0.853; 95% CI: 0.781–0.903; *p* < 0.0001). The HPLC chromatograms show high concentrations of CM-C in serum samples where the RemL-C level < the RLP-C level, but high concentrations of IDL-C in samples where the RemL-C level > the RLP-C level. RemL-C and RLP-C assays are likely to reflect IDL (VLDL remnant) and CM remnants, respectively.

### 3.1. Atherogenic Properties of Remnant Lipoproteins

The atherogenic properties of remnant lipoproteins are shown in Figure 2. Remnant lipoproteins have undergone extensive intravascular remodeling. Lipoprotein lipase (LPL), hepatic triglyceride lipase (HTGL), and cholesterol ester transfer protein (CETP) induce structural and atherogenic changes that distinguish remnant lipoproteins from non-remnant lipoproteins [77]. Via the rapid LPL-mediated removal of TGs and the CETP-mediated exchange of TGs for cholesterol from LDL and HDL, remnant particles contain more cholesterol than nascent CM or VLDL [78]. Compared with CM or VLDL, remnants lose apo CIII and become enriched in apo E [77]. While CM and VLDL are prohibited from transcytosis by virtue of their size, remnant lipoproteins can penetrate the artery wall (Figure 2(1)) [79,80,81]. Remnants efflux from the subendothelial space very slowly compared to native LDL and are therefore subject to increased internalization by macrophages (Figure 2(1)) [82,83]. Distinct from LDL, which is deficient in apo E and requires oxidation for uptake, remnants do not need oxidation to facilitate accumulation in macrophages due to their apo E enrichment.

VLDL remnant-treated monocytes show more rolling and adhesion than controls and show an increase in transmigration between endothelial cells (Figure 2(2)) [84]. The increased adhesive events are related to the elevated expression of key integrin complexes on treated monocytes. Treatment of peripheral blood mononuclear cells and monocytes with VLDL remnants increases the expression of tumor necrosis factor-alpha (TNF-α), interleukin-1beta (IL-1β), and IL-8 over controls (Figure 2(3)), with concurrent activation of nuclear factor kappa B (NF-kB) and activator protein 1 (AP-1) (Figure 2(4)). NF-κB- and AP-1-induced cytokine and integrin expression is dependent on ERK and Akt phosphorylation. VLDL remnants induce ERK2-dependent lipid droplet formation in monocytes, suggesting a link to inflammatory signaling pathways.

Remnant lipoprotein particles stimulate NAD(P)H oxidase-dependent superoxide formation and the induction of cytokines in endothelial cells via the activation of the lectin-like oxidized low-density lipoprotein receptor-1 (LOX-1) (Figure 2(5)), consequently leading to reductions in cell viability with DNA fragmentation [85]. Such oxidative stress increases the production of oxidized LDL (Figure 2(6)). Oxidized LDL induces the formation of foam cells via scavenger receptors.

Remnant lipoproteins from patients with type III hyperlipoproteinemia induce endothelial cell PAI-1 expression (Figure 2(7)), which may contribute to a prothrombotic state [86]. Remnant lipoproteins also induce platelet aggregation (Figure 2(8)) [87]. Mochizuki et al., investigated the in vitro influence of CM remnants and VLDL remnants on platelet aggregation in healthy persons [88]. Preincubation with CM and VLDL remnants significantly enhanced the platelet aggregation in whole blood and in platelet-rich plasma induced by collagen, indicating that an increase in remnant lipoproteins may contribute to atherosclerotic and thrombotic complications. The acute inflammatory response is associated with increased blood viscosity, platelet number, and activity [89].

### 3.2. Associations of Remnant Lipoproteins with CVD

Increased concentrations of remnant cholesterol were associated with increased all-cause mortality in patients with IHD, which was not the case for increased concentrations of LDL-C, suggesting that increased concentrations of remnant cholesterol explain part of the residual risk of all-cause mortality in patients with IHD [90].

Kaplan–Meier analysis demonstrated a significantly higher probability of developing coronary events in patients within the highest tertile of remnant levels (>5.1 mg/dL) than in those within the lowest tertile of remnant levels (≦3.3 mg/dL). Higher levels of remnants were a significant and independent predictor of developing coronary events in multivariate Cox hazard analysis, including the following covariates: the extent of coronary artery stenosis, age, sex, smoking, hypertension, diabetes, hypercholesterolemia, low HDL-C, and hypertriglyceridemia [91].

Serum levels of RLP-C were measured in 560 patients with coronary artery disease (CAD) who had LDL-C levels of <100 mg/dL on lipid-lowering therapy, including statin (58%), fibrate (13%), and diet only (29%) [92]. Stepwise multivariate Cox proportional hazard analysis showed that RLP-C was a significant predictor of CV events after adjustment for known risk factors and lipid variables including TGs, non-HDL-C, and total apo B (hazard ratio (HR): 1.53; 95% CI: 1.35–1.97; *p* < 0.01). RLP-C was superior to non-HDL-C for predicting CV events in CAD patients with LDL-C levels of <100 mg/dL on lipid-lowering treatment. Remnant lipoproteins may therefore be an important target for residual risk reduction after LDL-C goals in lipid-lowering therapy are met.

A total of 190 patients treated with statins after ACS were enrolled in a study [93]. Multivariate Cox analysis showed that a high level of RLP-C (≥5.4 mg/dL) was a significant risk factor for secondary events, independent of conventional risk factors (HR: 2.94; 95% CI: 1.40–6.18; *p* < 0.01).

Multivariate Cox analysis revealed that high levels of RLP-C (≥4.3 mg/dL) were a significant risk factor for CV events, independent of traditional risk factors (HR: 1.30; 95% CI: 1.04–1.63; *p* = 0.02), in patients with type 2 diabetes and CKD [94].

In a study of 4145 patients with type 2 diabetes, after multivariate logistic analyses, the level of plasma remnant cholesterol was significantly and independently associated with CAD (OR: 13.524; 95% CI: 7.058–25.912; *p* < 0.001) after adjustment for conventional risk factors, such as age, gender, hypertension, and other lipid levels [95].

A study was performed to examine the predictive value of remnant lipoprotein levels for CV events in patients with stable CAD and LDL-C levels of <70 mg/dL in response to statin treatment [96]. Kaplan–Meier analysis demonstrated that higher RLP-C levels (≥3.9 mg/dL) resulted in a significantly higher probability of a primary endpoint (cardiac death, nonfatal myocardial infarction, unstable angina requiring coronary revascularization, worsening heart failure, PAD, aortic event, or ischemic stroke) than did lower RLP-C levels (<3.9 mg/dL) (*p* < 0.01). Stepwise multivariate Cox proportional hazard analysis showed that RLP-C was a significant predictor of the primary endpoint after adjustment for known risk factors and lipid variables including TG levels and total apo B (HR: 1.62; 95% CI: 1.26–2.07; *p* < 0.01).

When values from 83 women with CVD were compared with the values from 1484 women without CVD, concentrations in women with CVD were found to be significantly higher for both RLP-C (+15.6%; *p* < 0.0001) and RLP-TG (+27.0%; *p* < 0.0002) [97]. Logistic regression analysis revealed that RLP-C was significantly associated with prevalent CVD in women (*p* < 0.002) after adjusting for other major risk factors.

## 4. Small Dense LDL (Sd-LDL)

A common lipoprotein profile designated with an atherogenic lipoprotein phenotype is characterized by a predominance of Sd-LDL. Multiple features of this phenotype, including increased levels of TG-rich lipoprotein remnants and IDLs, reduced levels of HDL, and an association with insulin resistance, contribute to an increased risk for CAD compared with individuals with a predominance of larger LDL [98].

### 4.1. Atherogenic Properties of Sd-LDL

The atherogenic properties of Sd-LDL are shown in Figure 3. Two major physically distinct species of VLDL exist: larger TG-rich VLDL1 (50–80 nm diameter; 70% TG mass) and smaller VLDL2 (30–50 nm diameter; 30% TG mass) [77]. At normal TG concentrations, VLDL1 and VLDL2 circulate in approximately equal proportions. Hepatic TG accumulation and insulin resistance increase VLDL1 secretion (Figure 3(1)) [99,100]. LDL derived from VLDL1 was also shown to have a slower clearance rate and longer duration to decay, inducing Sd-LDL formation [101,102]. An increased atherogenic potential of Sd-LDL is suggested by its greater propensity for transport into the subendothelial space (Figure 3(2)), increased binding to arterial proteoglycans (Figure 3(3)), and susceptibility to oxidative modification (Figure 3(4)) [98]. Sd-LDL is cleared more slowly from plasma than LDL due to its compositional and structural features (Figure 3(5)) [103].

The oxidative modification of LDL has been implicated in the pathogenesis of CAD. Macrophages take up oxidized LDL via scavenger receptors, which are not regulated by cellular cholesterol contents, and oxidized LDL stimulates cholesterol esterification, which results in cellular cholesterol accumulation and foam cell formation. Sd-LDL, which is LDL-poor in antioxidants (vitamin E, beta-carotene, and CoQ10), should be more susceptible to lipid peroxidation and possibly more atherogenic [104].

Three LDL subfractions, LDL1, LDL2, and LDL3, were isolated from the plasma of 11 healthy volunteers by density gradient ultracentrifugation. The study indicated that the denser LDL subfractions, that is, LDL2 and LDL3, are more susceptible to oxidative modification [105].

### 4.2. Associations of Sd-LDL with CVD

The association between large and small LDL and long-term IHD risk was investigated in men who participated in the Quebec Cardiovascular Study. This study confirmed the strong and independent association between Sd-LDL phenotype and the risk of IHD in men [106]. A prospective case-control study nested in the EPIC-Norfolk cohort was performed [107]. Cases involved apparently healthy men and women aged 45–79 years who had developed fatal or non-fatal CHD (*n* = 1035) and who were matched by age, gender, and enrollment time to 1920 controls who remained free of CHD. Sd-LDL levels were higher in cases than controls in men (1.34 ± 0.88 vs. 1.15 ± 0.80 mmol/L, *p* < 0.001) as well as in women (1.12 ± 0.84 vs. 0.94 ± 0.74 mmol/L, *p* < 0.001). The unadjusted OR for future CHD in men of the top tertile of LDL-C (<255 A) was 1.68 (95% CI: 1.33–2.13; *p* < 0.001), whereas the unadjusted OR was 1.53 in women (95% CI: 1.13–2.07; *p* < 0.001). Sd-LDL-C increased as the number of diseased vessels or the Gensini atherosclerosis score increased [108]. Among the 123 stable CHD patients, multiple logistic regression analysis revealed that Sd-LDL-C levels were significantly associated with clinically severe cases requiring coronary revascularization independently of LDL-C, HDL-C, and apo B. Both male and female CHD patients had significantly higher Sd-LDL-C concentrations than the control subjects [109]. Multivariate logistic regression analysis revealed that Sd-LDL-C levels were significantly associated with severe CHD, independent of LDL-C.

The relationship between plasma levels of Sd-LDL and risk for incident CHD in a prospective study among participants in the Atherosclerosis Risk in Communities (ARIC) study was investigated [110]. In a model that included established risk factors, Sd-LDL was associated with incident CHD with an HR of 1.51 (95% CI: 1.21–1.88) for the highest versus the lowest quartile. Even in individuals considered to be at low CV risk based on their LDL-C levels, Sd-LDL predicted the risk of incident CHD (HR: 1.61; 95% CI: 1.04–2.49).

A recent study showed that elevated levels of plasma Sd-LDL were associated with an increased risk of major cardiovascular events (MACEs) among diabetic patients with proven CAD (HR: 1.83; 95% CI: 1.24–2.70, *p* < 0.05) [111], suggesting that Sd-LDL may be useful for CAD risk stratification in patients with diabetes.

The presence of PAD was independently associated with elevated Sd-LDL (OR: 6.7; *p* = 0.0497) [112]. Patients with abdominal aortic aneurysm had increased levels of Sd-LDL (*p* = 0.0210) in relation to controls [113].

## 5. Therapeutic Approaches to Lower Remnant Lipoproteins and Sd-LDL

Elevation of serum remnant lipoproteins and Sd-LDL is induced by the abnormal metabolism of TG-rich lipoproteins due to insulin resistance [114,115]. Elevation of both remnant lipoproteins and Sd-LDL is commonly observed in insulin-resistant states such as obesity, metabolic syndrome, and type 2 diabetes. Therefore, in planning therapeutic strategies for remnant lipoproteins and Sd-LDL, it is thought that the same method can be used to treat both lipoproteins.

### 5.1. Diet

There are no reports on diet reducing remnant lipoproteins or Sd-LDL. Low HDL-C is observed in insulin resistance and is often accompanied with an elevation of TG-rich lipoproteins, remnant lipoproteins, and Sd-LDL. We previously performed a systematic review on diets that increase HDL-C and decrease TG [116]. To conduct the systematic review, we searched meta-analyses of RCTs that investigated the effects of energy and carbohydrate intake [117]; glycemic index (GI) and intake of dietary fiber [118]; intake of soy and non-soy legumes [119]; and consumption of various fatty acids (FA) [120] on serum HDL-C levels.

Low-carbohydrate diets may increase HDL-C, which may be due to reductions in body weight and/or the amelioration of insulin resistance [117]. Fructose intake may exert no effect on HDL-C; however, a fructose intake equal to or less than 100 g/day may be recommended, considering its unfavorable effects on TG levels. In spite of significant associations of low GI and dietary fiber intake with reductions in LDL-C, we could not observe any significant influences of low GI and dietary fiber intake on HDL and TG metabolism [118]. Intake of soy was significantly associated with elevations of HDL-C and reductions in TG, but non-soy legume consumption was not associated with a significant increase in HDL [119]. Consumption of ruminant-trans FA (TFA) may not affect HDL; however, increased industrially produced TFA intake was associated with a significant decrease in HDL [120]. Intake of n-3 poly-unsaturated FA (PUFA) and monounsaturated FA (MUFA) was associated with an increase in HDL and a decrease in TG. Dietary intervention with n-3 PUFA decreased fasting and postprandial TG levels, the number of remnant-like CM particles, large VLDL particles, and Sd-LDL particles [121].

### 5.2. Physical Activity

Physical training acts on both excessive VLDL production and the fractional catabolic rate of TG-rich lipoproteins [122]. A study was performed to examine the influence of moderate-intensity exercise on postprandial lipemia and muscle LPL activity [123]. Moderate exercise attenuates postprandial lipemia, possibly by altering muscle LPL activity. In one of our previous studies, supervised aerobic exercise training significantly decreased body mass index (BMI) and IDL-C and markedly reduced VLDL-C at weeks 8 (−45%) and 16 (−50%) [124].

Exercise induced reductions in VLDL1 concentrations in overweight/obese, middle-aged men, and these reductions were mediated by increased catabolism, rather than reduced production, which may have been facilitated by compositional changes to VLDL1 particles that increased their affinity for clearance from the circulation [125]. Exercise increases the affinity of VLDL1 for LPL-mediated TG hydrolysis in both fasting and postprandial states [126]. This mechanism is likely to contribute to the TG-lowering effect of exercise. Exercise training increased the VLDL1-TG and apo B fractional catabolic rates in nonalcoholic fatty liver disease (NAFLD) patients [127]. An increased clearance of VLDL1 may contribute to a significant decrease in liver fat after 16 weeks of exercise in NAFLD patients.

Exercise reduces VLDL by activating LPL, which may decrease remnant lipoproteins and Sd-LDL.

### 5.3. Statins

Patients with nephrotic-range proteinuria have an impaired clearance of TG-rich lipoproteins. A randomized crossover study compared the effects of a statin (cerivastatin) and a fibrate (fenofibrate) on Sd-LDL and RLP in patients with nephrotic-range proteinuria [128]. Cerivastatin reduced Sd-LDL concentration (27%; *p* < 0.01) but did not change RLP-C or RLP-TG.

Atorvastatin (10–20 mg/day) was administered for 3 months to 15 outpatients with hypercholesterolemia accompanied by hypertriglyceridemia without hypolipemic treatment [129]. Administration of atorvastatin significantly decreased Sd-LDL (*p* < 0.001) (before administration, 119 ± 17 mg/dL; after administration, 43 ± 10 mg/dL) and RLP-C (*p* < 0.01) (before administration, 11.9 ± 2.0 mg/dL; after administration, 6.0 ± 0.9 mg/dL). Atorvastatin reduced Sd-LDL from 34 ± 22% to 18 ± 20% (*p* < 0.01), and RLP-C reduced it from 8.8 ± 6.0 mg/mL to 5.1 ± 2.6 mg/mL (*p* < 0.01) in hypercholesterolemic patients [130]. Patients with type 2 diabetes are known to have abnormalities in their remnant metabolism and a preponderance of Sd-LDL. Pitavastatin significantly reduced RLP-C (−30.9 ± 30.5%) and reduced the proportion of Sd-LDL from 29.9 ± 26.2% to 19.7 ± 22.7% [131].

Statins, especially newly developed strong statins, can reduce remnant lipoproteins and Sd-LDL.

### 5.4. Fibrates

Fenofibrate lowered VLDL-C (52%), Sd-LDL (49%), RLP-C (35%), and RLP-TG (44%) (all *p* < 0.01) in patients with nephrotic-range proteinuria [127]. Twenty hypertriglyceridemic men were administered fenofibrate, 200 mg daily, for 8 weeks [128]. VLDL, particularly large VLDL, IDL (VLDL remnant), and small LDL were significantly decreased [132]. Bezafibrate lowered large TG-rich lipoproteins and IDL (VLDL remnant) by 81% and 46%, respectively [133]. Fenofibrate treatment resulted in significant (*p* < 0.05) changes versus a placebo in VLDL-C (−32.7%), RLP-C (−35.1%), and Sd-LDL (−35 vs. 21 mg/dL) [134].

### 5.5. Ezetimibe

Patients with type IIb, or mixed, dyslipidemia have high levels of LDL-C with a predominance of Sd-LDL. The efficacy and safety of the coadministration of fenofibrate and ezetimibe in patients with type IIb dyslipidemia and metabolic syndrome has been compared with those of the administration of fenofibrate and ezetimibe alone. Fenofibrate and ezetimibe was as effective as fenofibrate alone and more effective than ezetimibe alone in reducing RLP-C (−36.2% and −30.7% vs. −17.3%, respectively) and in increasing LDL size (+2.1% and +1.9% vs. + 0.7%, respectively) [135]. Ezetimibe reduced RLP-C (−22%, *p* < 0.001) and Sd-LDL (−19%, *p* < 0.001) in patients with diabetes and glucose intolerance [136].

Statins are used to treat hypercholesterolemia in patients with type 2 diabetes, but many of these patients fail to achieve the target LDL-C level. Recent reports have suggested that a synergistic effect can be obtained by the concomitant administration of the cholesterol absorption inhibitor ezetimibe and a statin. Type 2 diabetic patients under treatment with rosuvastatin (2.5 mg daily) who had LDL-C levels of ≥80 mg/dL were randomly allocated to a group that received add-on therapy with ezetimibe at 10 mg/day (combination group, *n* = 40) or an increase of the rosuvastatin dose to 5 mg/day (dose escalation group, *n* = 39) [137]. In both groups, there was a significant decrease in the levels of Sd-LDL-C and RLP-C. For all of these parameters, the percent changes were greater in the combination group.

Although dual LDL-C-lowering therapy (DLLT) with a statin–ezetimibe combination showed clinical benefit in patients with ACS, confirming “the lower, the better”, the underlying mechanisms of DLLT are still unknown. Lower RLP-C and stronger reductions in Sd-LDL-C were observed in patients with plaque regression compared to those without progression [138].

## 6. Malondialdehyde-Modified Low-Density Lipoprotein (MDA-LDL)

In recent years, oxidized LDL has attracted attention as a blood marker that is associated with CAD. Oxidized LDL plays important roles in the formation and development of the primary lesions of atherosclerosis. An ELISA technique was developed for the measurement of malondialdehyde-modified low-density lipoprotein (MDA-LDL), a representative form of oxidized LDL [139]. MDA-LDL levels were higher in CAD patients [140] and diabetic patients [141] than in control subjects.

### 6.1. Associations of MDA-LDL with CVD

Plasma levels of MDA-LDL were significantly higher in patients with ACS than in individuals with stable CAD (*p* = 0.0001), suggesting that elevated plasma levels of MDA-LDL suggest plaque instability and may be useful for the identification of patients with ACS [142]. MDA-LDL discriminated between stable CAD and unstable angina (*p* = 0.001) [143]. The sensitivity of MDA-LDL was 95% for unstable angina and 95% for acute myocardial infarction, with a specificity of 95%. MDA-LDL levels in coronary circulation are associated with the development of atherothrombosis from the progression of atherosclerosis rather than with the extent of coronary atherosclerosis in patients with coronary spastic angina [144]. The MDA-LDL level was increased in patients with CAD, independent of the serum LDL-C level, and was negatively correlated with the peak size of the LDL particle (*p* < 0.01) [141]. The measurement of serum MDA-LDL level may be useful for the identification of patients with advanced atherosclerosis. The MDA-LDL levels were significantly higher in the CAD group than in the control group (*p* = 0.01), even though there was no significant difference in the LDL-C levels between the two groups [145]. Nuclear magnetic resonance (NMR) analysis demonstrated that the MDA-LDL levels were positively correlated with large and intermediate VLDL-TG and LDL particle concentrations and negatively correlated with LDL diameter and large HDL-C. The MDA-LDL levels were negatively correlated with flow-mediated dilatation of the brachial artery, suggesting that circulating MDA-LDL may impair endothelial functions and play an important role in the pathogenesis of atherosclerosis.

An elevated serum MDA-LDL level is considered to be a potent risk factor for in-stent restenosis (ISR) after intracoronary stenting in diabetic patients [146]. MDA-LDL, which might be a consequence of metabolic abnormalities caused by diabetes, may act as a growth factor for neointimal tissues inside the implanted stent. The relationship between coronary plaque vulnerability assessed by optical coherence tomography (OCT) and circulating MDA-LDL was studied [147]. Thin-cap fibroatheromas (TCFAs; defined as lipid-rich with a plaque cap thickness of <65 μm) were detected more frequently in ACS than stable angina pectoris (SAP) (83% vs. 16%, *p* < 0.001). MDA-LDL levels were significantly higher in patients with ACS compared with SAP (*p* = 0.008). The levels of MDA-LDL were significantly higher in SAP patients with TCFA than in non-TCFA patients (*p* < 0.001). Patients with ruptured TCFA had higher levels of MDA-LDL compared with those with morphologically intact TCFA (*p* = 0.023). Circulating MDA-LDL levels might be associated with the presence of TCFA in the culprit lesion. Another study also suggested that elevated MDA-LDL was confirmed to be associated with thin-cap atheroma in CAD patients [148], supporting a significant association of MDA-LDL with plaque vulnerability. MDA-LDL above the 75th percentile is a marker of LDL oxidation that predicts a worse CV prognosis at long-term follow-up in high-risk Caucasian patients referred for coronary angiography [149].

### 6.2. Therapeutic Approaches to Lower MDA-LDL

#### 6.2.1. Statins

A study was performed to compare the effects of atorvastatin and pravastatin on MDA-LDL in hypercholesterolemic patients [150]. The percent reductions in LDL-C and MDA-LDL concentration were significantly greater with atorvastatin than pravastatin (46 ± 6% vs. 24 ± 0%, *p* < 0.0001, and 44 ± 10% vs. 14 ± 13%, *p* < 0.0001, respectively). In a six-month prospective study, 75 patients with stable CAD were randomly assigned to a pravastatin treatment group (*n* = 52) or a control group (*n* = 23) [151]. Pravastatin therapy for 6 months resulted in a decrease in coronary plaque volume (14.4%, *p* < 0.0001) and a corresponding reduction in serum MDA-LDL levels (12.7%, *p* = 0.0001). In the pravastatin treatment group, the percentage of change in plaque volume correlated with changes in the MDA-LDL level (r = 0.52, *p* < 0.0001). Multivariate regression analysis revealed that a reduced MDA-LDL level is an independent predictor of plaque regression.

In a prospective, randomized, open-label, multicenter study, a total of 111 high-risk patients were randomly assigned to two groups [152]. In the high-dose therapy group, 58 patients were administered 5 mg of rosuvastatin per day for four weeks, after which the dose was titrated to 10 mg for the following eight weeks. Sd-LDL and MDA-LDL levels were significantly reduced in this group (*p* < 0.05).

#### 6.2.2. Fibrates

The MDA-LDL/apo B ratio was lower in a fibrate group compared with a no-drug group (*p* < 0.01) in diabetic patients [153].

#### 6.2.3. Ezetimibe

Ezetimibe treatment for 12 weeks significantly reduced MDA-LDL in hypercholesterolemic patients; ezetimibe therapy resulted in a decrease in MDA-LDL levels by 27% [154]. Another study also showed that ezetimibe monotherapy significantly reduced serum levels of MDA-LDL [155]. Subjects with CAD and type 2 diabetes or impaired glucose tolerance who were taking 10 mg/day of atorvastatin were randomized to a group that first received add-on ezetimibe (10 mg/day) or a group that first received atorvastatin monotherapy at a higher dose of 20 mg/day. Add-on ezetimibe significantly decreased MDA-LDL from 109.0 ± 31.9 mg/dL to 87.7 ± 29.4 mg/dL (*p* = 0.0009), while the up-titration of atorvastatin did not [156]. Ezetimibe reduced the mean levels of MDA-LDL (−15%, *p* < 0.001) in patients with diabetes and glucose intolerance [136].

To understand whether the addition of ezetimibe to statin therapy is more effective than double-dose statin monotherapy, a crossover design study was conducted. Twenty-one CAD patients whose lipid profiles had not achieved Japanese guideline recommendations, despite receiving low-dose statin therapy, were divided into two groups. Group A received 10 mg of ezetimibe in addition to a baseline dose of statin for the first three months and was then switched to monotherapy with a double dose of statin for the next three months. Group B first received a double dose of statin for three months and was then switched to 10 mg of ezetimibe in addition to a baseline dose of statin for the next three months. Compared with the baseline, double-dose statin therapy reduced MDA-LDL from 142 ± 35 U/L to 126 ± 24 U/L (*p* < 0.05). The addition of ezetimibe to a baseline dose of statins further reduced MDA-LDL to 114 ± 22 U/L (*p* < 0.001) [157].

## 7. Factors Other Than Atherogenic Lipoproteins in Statin Residual CVD Risk

One of our previous studies demonstrated that the presence of obesity/insulin resistance and diabetes may be important metabolic factors that determine statin residual CVD risk [12]. Insulin resistance is frequently associated with visceral obesity, and the features of low-grade inflammation, including elevated levels of C-reactive protein and interleukin-6, have been associated with visceral obesity [158]. Adipose tissue generation of cytokines has been shown in vitro and in vivo, and many novel cytokine-like molecules, collectively termed adipocytokines, have been identified as adipocyte products [158]. Chronic inflammation is a pathogenic feature of atherosclerosis [159]. Direct injury to a vessel wall causes endothelial and smooth muscle cells of large arteries to become transcriptionally active and synthesize pro-inflammatory proteins, including chemokines, cell adhesion molecules, and cytokines as well as growth factors and prothrombogenic substances. Cytokine-activated macrophages and smooth muscle cells secrete matrix metalloproteinases, which, when activated, digest connective tissue elements within the vessel wall and thin the fibrous cap overlying vulnerable plaques. In patients with CVD, elevated serum interleukin-6 levels have been linked to excessive CVD outcomes and death [160,161,162]. In addition to these studies, many others have demonstrated a significant contribution of inflammation to CVD development. Inflammation can play a crucial role in statin residual CVD risk.

## 8. Summary

To summarize our review, determinants of serum concentration, indicators in clinical practice, and recommended therapeutic approaches for elevation of lipoprotein (a), remnant lipoproteins, small-dense LDL, and MDA-LDL are shown in Table 1.

## 9. Conclusions

We showed the atherogenic properties of Lp(a), remnant lipoproteins, MDA-LDL, and Sd-LDL, the association of such lipoproteins with CVD development, and therapeutic approaches for them. The measurements of such lipoproteins and appropriate therapeutic interventions are crucial to reduce statin residual CVD risk.

## Figures and Tables

**Figure 1 ijms-23-13499-f001:**
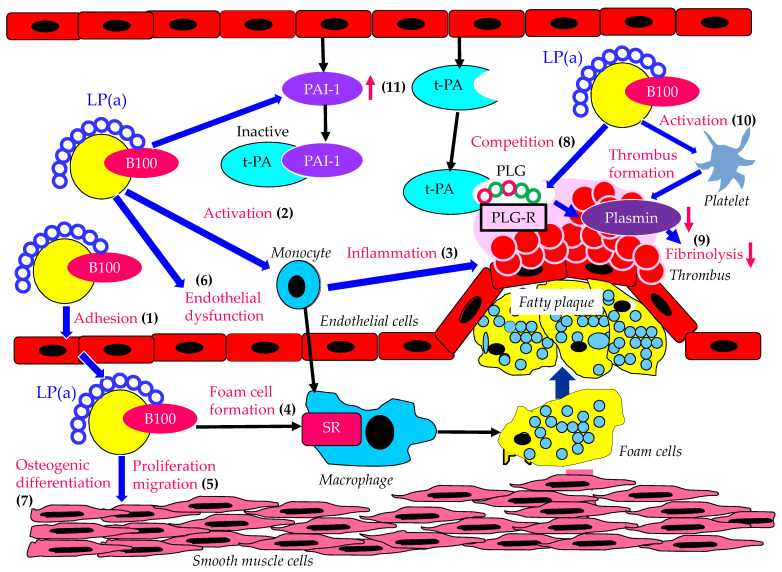
Atherogenic and thrombogenic properties of Lp(a). Lp(a): lipoprotein (a); PAI-1: plasminogen activator inhibitor-1; PLG: plasminogen; PLG-R: plasminogen receptors; SR: scavenger receptor; t-PA: tissue plasminogen activator.

**Figure 2 ijms-23-13499-f002:**
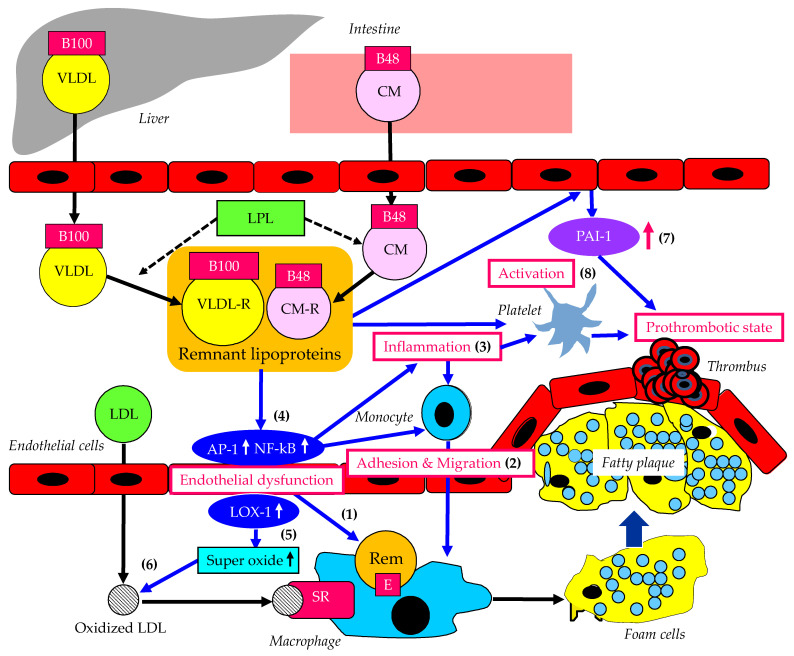
Atherogenic properties of remnant lipoproteins. AP-1: activator protein 1; CM: chylomicron; CM-R: chylomicron remnant; LDL: low-density lipoprotein; LOX-1: lectin-like oxidized low-density lipoprotein receptor-1; NF-kB: nuclear factor kappa B; PAI-1: plasminogen activator inhibitor-1; Rem: remnant lipoproteins; SR: scavenger receptor; VLDL: very-low-density lipoprotein; VLDL-R: very-low-density lipoprotein remnant.

**Figure 3 ijms-23-13499-f003:**
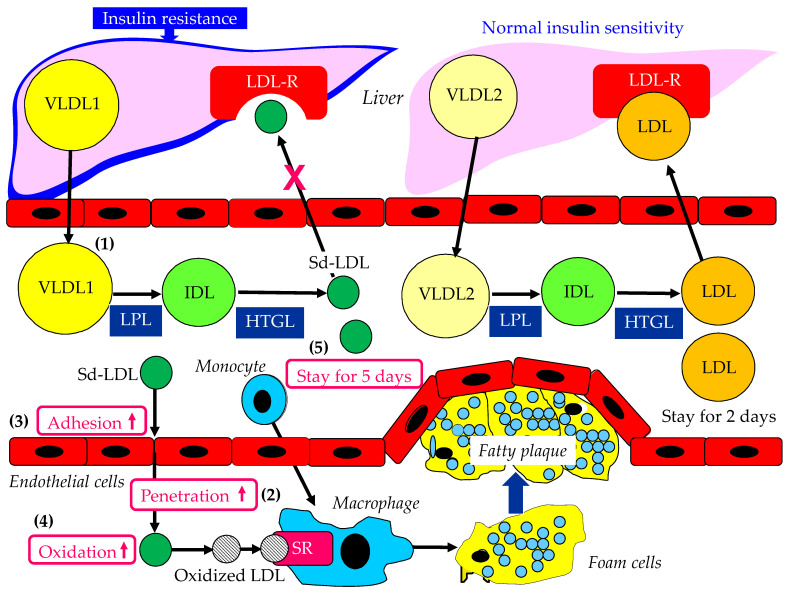
Atherogenic properties of Sd-LDL. Red cross indicates the reduced LDL receptor affinity of sd-LDL. HTGL: hepatic triglyceride lipase; IDL: intermediate-density lipoprotein; LDL: low-density lipoprotein; LDL-R: low-density lipoprotein receptor; LPL: lipoprotein lipase; Sd-LDL: small dense low-density lipoprotein; SR: scavenger receptor; VLDL: very-low-density lipoprotein.

**Table 1 ijms-23-13499-t001:** Determinants of serum concentration, indicators in clinical practice, and recommended therapeutic approaches for elevation of lipoprotein (a), remnant lipoproteins, small-dense LDL, and MDA-LDL.

	Lipoprotein (a)	Remnant Lipoproteins	Small Dense LDL	MDA-LDL
Determinants of serum concentration	Genetic factors (91%)	Environmental factors (type 2 diabetes, metabolic syndrome, chronic kidney disease) and genetic factors (familial combined hyperlipidemia, type III hyperlipidemia)	Environmental factors (almost the same as remnant lipoproteins)	Environmental factors (coronary artery disease, type 2 diabetes)
Indicators in clinical practice	None	Obesity	Obesity	Elevated LDL
		Insulin resistance	Insulin resistance	Diabetes
		Hypertriglyceridemia	Hypertriglyceridemia	
		Reduced HDL	Reduced HDL	Coronary artery disease
Recommended Therapeutic approach	Niacin (−18~23%)	Exercise	Exercise	Statin (−12~44%)
	Hormone replacement therapy (−20~25%)	Diet therapy	Diet therapy	
	PCSK9 inhibitors (−20~30%)	Body weight reduction	Body weight reduction	Fibrates
		Statin (−30~50%)	Statin (−34~64%)	
	ASOs (−50~80%) and siRNA targeting Lp(a)	Fibrates (−35~46%)	Fibrates (−35~49%)	Ezetimibe (−15~27%)
		Ezetimibe (−17~22%)	Ezetimibe (−19%)	

ASOs, antisense oligonucleotides; HDL, high-density lipoprotein; MDA-LDL, malondialdehyde-modified LDL; LDL, low-density lipoprotein; PCSK9, proprotein convertase subtilisin/kexin type 9; siRNA, small interfering RNA.

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
