# Peer review of "Atherogenic Lipoproteins for the Statin Residual Cardiovascular Disease Risk"

_ijms, 2022, doi:10.3390/ijms232113499_

Round 1
Reviewer 1 Report
Inasmuch as cardiovascular disease (CVD) accounts for approximately 32% of deaths worldwide, the paper under Review explores an area of high interest to clinicians and patients alike. Statin-based therapeutic interventions have figured highly among the standards of care for decades. Accordingly, the subject matter discussing potential weak spots in the approach represents a significant realm for critical analysis.
Here, and not without merit, the Authors focus much of their manuscript upon two facets of a key precursor of therapeutic escape - i.e. Remnant lipoproteins and MDA-LDL - for example, observing that MDA-LDL is a canonical form of oxidized LDL which contributes to the formation and development of the primary atherosclerotic lesions.
At this point the Authors go into rather extensive detail describing a series of diverse phenotypic expressions wherein lipoprotein A (Lpa) contributes to onset and progression of disease. Here, the Reviewer finds a key missing variable. Although briefly touched upon, e.g. figures 1 & 2, the underlying mechanisms governing inflammatory sequelae associated with CVD (both chronic & acute) appear to be largely absent.
A cursory pubmed.com search of "interluekins AND cardiovascular disease" returned over 32,000 hits. Likewise, a similar, but more focused search with the keywords "inflammatory precursors AND cardiovascular disease" returned more than 1,000 hits. One may therefore ponder the dearth of discussion in this paper addressing such axiomatic variables.
Likewise, the genetic underpinnings of LPa mediated cardiovascular disease are only very lightly touched upon - though current state-of-the-art therapeutic interventions now tend to rely upon useful apprehension of such data points. Most importantly, many of these variants have been shown to confer a manifold increase in risk for coronary artery disease, myocardial infarction, and ventricular fibrillation.
Finally, to be easily accessible to the target audience, a thorough and detailed English language editing of this Paper must occur.
Author Response
Dear Reviewer 1
1. According to the suggestion “Here, the Reviewer finds a key missing variable. Although briefly touched upon, e.g. figures 1 & 2, the underlying mechanisms governing inflammatory sequelae associated with CVD (both chronic & acute) appear to be largely absent. A cursory pubmed.com search of "interluekins AND cardiovascular disease" returned over 32,000 hits. Likewise, a similar, but more focused search with the keywords "inflammatory precursors AND cardiovascular disease" returned more than 1,000 hits. One may therefore ponder the dearth of discussion in this paper addressing such axiomatic variables.”
We added the following sentences by citing ref. 159-163.
7. The factor other than atherogenic lipoproteins in the statin residual CVD risk
Our previous study demonstrated that the existence of obesity/insulin resistance and diabetes may be important metabolic factors which determine the statin residual CVD risk [12]. Insulin resistance is frequently associated with visceral obesity, and the features of low-grade inflammation, including elevated levels of C-reactive protein and interleukin-6, have been associated with visceral obesity [159]. Adipose tissue generation of cytokines has been shown in vitro and in vivo, and a number of novel cytokine-like molecules, collectively termed adipocytokines, have been identified as adipocyte products [159]. Chronic inflammation is a pathogenic feature of atherosclerosis [160]. Direct injury to the vessel wall causes endothelial and smooth muscle cells of large arteries to become transcriptionally active and synthesize pro-inflammatory proteins, including chemokines, cell adhesion molecules, and cytokines as well as growth factors and prothrombogenic substances. Cytokine-activated macrophages and smooth muscle cells secrete matrix metalloproteinases, which, when activated, digest connective tissue elements within the vessel wall and thin the fibrous cap overlying vulnerable plaques. In patients with CVD, elevated serum interleukin-6 levels have been linked to excessive CVD outcomes and death events [161-163]. Besides these studies. a great number of studies have demonstrated a significant contribution of inflammation to the development of CVD. Inflammation can play a crucial role in the statin residual CVD risk.
2. According to the suggestion “Likewise, the genetic underpinnings of LPa mediated cardiovascular disease are only very lightly touched upon - though current state-of-the-art therapeutic interventions now tend to rely upon useful apprehension of such data points. Most importantly, many of these variants have been shown to confer a manifold increase in risk for coronary artery disease, myocardial infarction, and ventricular fibrillation.”
We added the following sentences.
The variants in LPA have been shown to confer a significant increase in risk for CHD [33, 34], ischemic stroke subtype large artery atherosclerosis, PAD, and abdominal aortic aneurysm [42], aortic valve calcification [46] and aortic valve stenosis [47]. The current state-of-the-art therapeutic interventions now tend to rely upon useful apprehension of such data points. A meta-analysis showed that the variants in LPA are associated with CVD events during statin therapy independently of the extent of LDL-lowering, suggesting a significant contribution of LPA variants to the statin residual CVD risk [51].
3. According to the suggestion “Finally, to be easily accessible to the target audience, a thorough and detailed English language editing of this Paper must occur.”
We asked the MDPI editing service to edit and correct our MS.
We deeply appreciate your kind and scientific advice.
Reviewer 2 Report
This review paper summarized despite drug treatment, there is still a risk of developing CVD. The reason is thought to be that the lipid metabolism process is affected, such as obesity, diabetes, etc..., which elevated lipoprotein(a) [Lp(a)], residual lipoprotein, malondialdehyde-modified low-density lipoprotein (MDA-LDL), and small-density low-density lipoprotein (Sd-LDL). The author of this article discusses the above factors and gives treatment recommendations.
Specific comments
1. How to detect the above indicators in clinical practice? And whether these indicators are generally seen.
2. Are there any commonalities or differences among these factors?
3. In 2.3.8, Has targeted therapy to lower LP(a) entered the third phase?
4. It is recommended to place the process sequence number in the figure for the convenience of readers
5. It may be possible to organize the whole text through tables to make the full text more complete.
6. In the conclusions, need to explore the proposed abnormal lipoprotein and the incidence of cardiovascular disease left by statin, the author thinks how to treat it better?
Author Response
Dear Reviewer 2
1. According to the suggestion “ How to detect the above indicators in clinical practice? And whether these indicators are generally seen. 2. Are there any commonalities or differences among these factors? 5. It may be possible to organize the whole text through tables to make the full text more complete. 6. In the conclusions, need to explore the proposed abnormal lipoprotein and the incidence of cardiovascular disease left by statin, the author thinks how to treat it better?”
We made Table 1 and added the following section.
8. Summary
To summarize our review, determinants of serum concentration, indicators in clinical practice and recommended therapeutic approach for elevation of lipoprotein (a), remnant lipoproteins, small-dense LDL and MDA-LDL were shown in Table 1.
| Table 1. Determinants of serum concentration, indicators in clinical practice and recommended therapeutic approach for elevation of lipoprotein (a), remnant lipoproteins, small-dense LDL and MDA-LDL. | ||||
|   | Lipoprotein (a) | Remnant lipoproteins | Small dense LDL | MDA-LDL |
| Determinants of serum concentration | Genetic factors (91%) | Environmental factors (type 2 diabetes, metabolic syndrome, chronic kidney disease) and Genetic factors (familial combined hyperlipidemia, type III hyperlipidemia) | Environmental factors (almost same as remnant lipoproteins) | Environmental factors (coronary artery disease, type 2 diabetes) |
| Indicators in clinical practice | None | Obesity Insulin resistance Hypertriglyceridemia Reduced HDL | Obesity Insulin resistance Hypertriglyceridemia Reduced HDL | Elevated LDL Diabetes Coronary artery disease |
| Recommended Therapeutic approach | Niacin (-18~23%) Hormone replacement therapy (-20~25%) PCSK9 inhibitors (-20~30%) ASOs (-50~80%) and siRNA targeting Lp(a) | Exercise Diet therapy Body weight reduction Statin (-30~50%) Fibrates (-35~46%) Ezetimibe (-17~22%) | Exercise Diet therapy Body weight reduction Statin (-34~64%) Fibrates (-35~49%) Ezetimibe (-19%) | Statin (-12~44%) Fibrates Ezetimibe (-15~27%) |
| ASOs, antisense oligonucleotides; HDL, high-density lipoprotein; MDA-LDL, malondialdehyde-modified LDL; LDL, low-density lipoprotein; PCSK9, proprotein convertase subtilisin/kexin type 9; siRNA, small interfering RNA. | ||||
2. According the suggestion “ It is recommended to place the process sequence number in the figure for the convenience of readers”
We added the process sequence number in the figures and text.